Predictions of potential geographical distribution and quality of Schisandra sphenanthera under climate change

Guo Yanlong 1 2 3
Wei Haiyan weihy@snnu.edu.cn 3
Lu Chunyan 4
Gao Bei 1 3
Gu Wei weigu@snnu.edu.cn 1 5
1 National Engineering laboratory for Resource Development of Endangered Chinese Crude Drugs in Northwest of China, Shaanxi Normal University , Xian , China
2 Cold and Arid Regions Environmental and Engineering Research Institute, Chinese Academy of Sciences , Lanzhou , China
3 College of Tourism and Environment, Shaanxi Normal University , Xian , China
4 Fujian Agriculture and Forestry University, College of Computer and Information Sciences , Fuzhou , China
5 College of Life Sciences, Shaanxi Normal University , Xian , China
Miao Chiyuan
Electronic publication date: 2016 Oct 20
Publication date: 2016
Volume: 4
Electronic Location ID: e2554
Received 2016 Mar 18; Accepted 2016 Sep 12
Copyright: ©2016 Guo et al.
Copyright year: 2016
Copyright holder: Guo et al.
License: This is an open access article distributed under the terms of the Creative Commons Attribution License, which permits unrestricted use, distribution, reproduction and adaptation in any medium and for any purpose provided that it is properly attributed. For attribution, the original author(s), title, publication source (PeerJ) and either DOI or URL of the article must be cited.
License URL: https://creativecommons.org/licenses/by/4.0/

Keywords: Climate change, GIS, Medicinal plants, Fuzzy membership function, Schisandra sphenanthera, Species distribution modeling

Funding: National Natural Science Foundation of China No.31070293 Ministry of Science and the Technology of the People’s Republic of China No.2006BAI06A13-06 This work is supported by the National Natural Science Foundation of China (No.31070293), the National Eleventh-Five Year Science and Technology Support Program from Ministry of Science and the Technology of the People’s Republic of China (No.2006BAI06A13-06). The funders had no role in study design, data collection and analysis, decision to publish, or preparation of the manuscript.

==============================
Climate change will significantly affect plant distribution as well as the quality of medicinal plants. Although numerous studies have analyzed the effect of climate change on future habitats of plants through species distribution models (SDMs), few of them have incorporated the change of effective content of medicinal plants. Schisandra sphenanthera Rehd. et Wils. is an endangered traditional Chinese medical plant which is mainly located in the Qinling Mountains. Combining fuzzy theory and a maximum entropy model, we obtained current spatial distribution of quality assessment for S. spenanthera. Moreover, the future quality and distribution of S. spenanthera were also projected for the periods 2020s, 2050s and 2080s under three different climate change scenarios (SRES-A1B, SRES-A2 and SRES-B1 emission scenarios) described in the Special Report on Emissions Scenarios (SRES) of IPCC (Intergovernmental Panel on Climate Change). The results showed that the moderately suitable habitat of S. sphenanthera under all climate change scenarios remained relatively stable in the study area. The highly suitable habitat of S. sphenanthera would gradually decrease in the future and a higher decline rate of the highly suitable habitat area would occur under climate change scenarios SRES-A1B and SRES-A2. The result suggested that in the study area, there would be no more highly suitable habitat areas for S. sphenanthera when the annual mean temperature exceeds 20 °C or its annual precipitation exceeds 1,200 mm. Our results will be influential in the future ecological conservation and management of S. sphenanthera and can be taken as a reference for habitat suitability assessment research for other medicinal plants.

Introduction

Climate is the main determining factor in the distribution of species, and the change of species distribution patterns can be reflected by climate change definitely and directly (Root et al., 2003; Parmesan & Yohe, 2003; Lenoir et al., 2008; Bertrand et al., 2011; Bystriakova, Peregrym & Dragicevic, 2015). Global warming is likely to change the structure and function of essential terrestrial ecosystems (Wu et al., 2010; Euskirchen, Carman & McGuire, 2014) by changing the range of habitats and the distribution area, which has strong effects on herb spatial patterns and can increase the risk of extinction among endangered plants (Bartholomeus et al., 2011; Li et al., 2013). Recent studies have focused on the potential impact of climate change on vegetation distribution and nature reserves (Araújo et al., 2011; Zhang et al., 2014) and have raised great concerns about the future of a range of animal and plant species (Araújo & Rahbek, 2006; Fan et al., 2014; Guo et al., 2014). Although there are an increasing number of studies on the effect of climate change on the spatial distribution of plant species, only a few of them refer to endangered geo-herbs, and studies about the impact of climate change on herb quality are even fewer.

Medicinal plants play a very important role in health care, particularly in developing countries (Madaleno, 2010; Williamson et al., 2013). At present, many medicinal plant species are severely threatened by over exploitation, destructive harvesting, deforestation and habitat deterioration (Guo et al., 2014). The production and quality of medicinal plants are not only influenced by plant genetic characteristics but are also closely related to various environmental factors. Specific geographic spatial information about species is an important part of species conservation (Lu et al., 2012; Guo et al., 2013). This information is essential to addressing the many challenges facing species conservation, such as those induced by climate change, as well as other ecological or biological factors. It is also important for the domestication of wild species.

Schisandra sphenanthera (S. sphenanthera) is an important Chinese herbal medicine (Smith, 1947; Committee of Flora of China, 1996). Currently, it has been included in the Chinese Key Protected Wild Medicinal Species List, and it is considered to be a severely reduced resource among the major common species of wild medicinal herbs (Lu et al., 2012; Guo et al., 2013). Without enough knowledge about the habitat requirements of this species, species resource protection is improbable.

Models are essential tools to assess the potential response of vegetation to climate change, particularly if large spatial and temporal scales are considered (Araújo & Luoto, 2007; Austin & Van Niel, 2011; Adhikari, Barik & Upadhaya, 2012; Bean et al., 2014). SDMs are increasingly used to forecast the potential changes in species distributions under climate change scenarios. However, traditional SDMs can neither predict the change of plants’ medicinal component contents in different geographical conditions nor the potential response of plants’ medicinal component contents to climate change (Lu et al., 2012; Guo et al., 2013).

In this study, researchers collected S. sphenanthera samples at 307 sampling points in 19 sampling sites and extracted schisantherin A using high performance liquid chromatography (HPLC). Meanwhile, in order to test applicability of our model for other habitats besides the Qinling Mountains, 60 samples from four locations have been collected. We used a fuzzy set to determine the association of schisantherin A content with the 19 climatic variables. Then, the maximum entropy model was used to determine the weight of each variable. Finally, we estimated the spatial distribution of S. sphenanthera in the Qinling Mountains using the weighted average method and geographical information system (GIS) spatial analysis. Researchers obtained the suitable range of each variable affecting plant growth and spatial distribution. By using this habitat-suitability assessment model and the same climatic data from three general circulation models (GCM) for three different climate change scenarios (Special Report On Emissions Scenarios, SRES) (SRES-A1B, SRES-A2 and SRES-B1 emission scenarios), the researchers predicted the potential geographic distribution of S. sphenanthera in the Qinling Mountains for the years 2020–2029 (2020s), 2050–2059 (2050s) and 2080–2089 (2080s). This information will supply construction and advice for the protection and sustainable utilization of resources, and provide some measures and proposals for the prediction of the distribution and quality of medicinal plants under climate change conditions.

Materials and Methods

Study area and species data

In this study, we selected the Qinling Mountains as our study area. The Qinling Mountains are one of the main producing areas of medicinal materials and plants. And this area also contains a large number of endangered geoherbs (Dang et al., 2010; Lu et al., 2012). Climate change will certainly affect the Qinling Mountains climate, which will inevitably affect the habitable environment and moisture and temperature conditions in the Qinling Mountains (Fan et al., 2014), thus affecting herb spatial patterns and quality in the region.

According to the literature record of the distribution of S. sphenanthera and studies on the range of the Qinling Mountains (Liu & Chen, 1986; Kang & Zhu, 2007), we defined the whole study area to encompass the Qinling Mountains (Fig. 1). Latitude in this area ranges from N30 °30′ to N35 °30′ and longitude from E103 °45′ to E113 °45′. It runs across six provinces (or municipalities) from the west to the east, including Gansu, Sichuan, Shaanxi, Chongqing, Hubei, and Henan, and it covers nearly 190,000 km2.

Figure 1 Study area and sampling sites distribution.

a Huating, in Gansu; b Zhouqu, in Gansu; c Ningqiang, in Shaanxi; d Fengxian, in Shaanxi; e Longxian, in Shaanxi; f Liuba, in Shaanxi; g Maoping, in Shaanxi; h Foping, in Shaanxi; i Taibai, in Shaanxi; j Ningshan, in Shaanxi; k Yingpan, in Shaanxi; l Zhen’an, in Shaanxi; m Fengzhen, in Shaanxi; n Zhenping, in Shaanxi; o Xunyang, in Shaanxi; p Huaxian, in Shaanxi; q Qingchuan, in Sichuan; r Wuxi, in Chongqing; s Lushi, in Henan. 1 Lingchuan, in Shanxi; 2 Xiuwu, in Henan; 3 Jinzhai, in Anhui; 4 Lin’an, in Zhejiang.

We collected S. sphenanthera samples throughout the range of the Qinling Mountains from 2007–2011 (Fig. 1). We chose the reasonable sampling sites by considering the different environmental conditions in traditional and original producing area of S. sphenanthera. Meanwhile, to improve the sampling accuracy, the sampling points were distributed evenly. But because of the influence of human activities, S. sphenanthera wild resources are being depleted, and some traditional areas where the plant could be found no longer host this species, at the same time in complex terrain, we are unable to obtain sample point data, so the spatial distribution of the sampling points in this study are not completely evenly distribution. At every sampling point, 10–20 plant samples had been collected, and to prevent cloning, the distance between samples was controlled more than 100 m. We used global positioning system (GPS) to record basic geographic information, including the longitude, latitude, altitude, slope and aspect. The Committee of National Pharmacopoeia (2010) has clearly defined that schisantherin A is a symbol component in the fruit of S. sphenanthera. Therefore, the content of schisantherin A in fruit of S. sphenanthera was chosen as the indicator, and it was tested by HPLC (Gu, Wei & Wang, 2008; Lu et al., 2012).

Climatic variables, future scenarios and spatial conversion

In order to work out the geographic distribution of suitable habitat of a target species, a set of characteristics about this species must be defined (Lu et al., 2012). In this study,we used 19 bioclimatic variables (Table 1) (Nix, 1986). And we obtained bioclimatic variables from the WorldClim database (www.worldclim.org/current). These variables were generated using averaged interpolated climate data during the period from 1950 to 2000 (Hijmans et al., 2005), and the resolution was 30” (about 1 km2).

Table 1 Explanatory variables used to model of the distribution for S. sphenanthera.

Code	Description	
Bio1	Annual mean temperature	
Bio2	Mean diurnal temperature range	
Bio3	Isothermality (BIO2/BIO7) (* 100)	
Bio4	Temperature seasonality (standard deviation *100)	
Bio5	Max temperature of warmest month	
Bio6	Min temperature of coldest month	
Bio7	Temperature annual range (BIO5-BIO6)	
Bio8	Mean temperature of wettest quarter	
Bio9	Mean temperature of driest quarter	
Bio10	Mean temperature of warmest quarter	
Bio11	Mean temperature of coldest quarter	
Bio12	Annual precipitation	
Bio13	Precipitation of wettest month	
Bio14	Precipitation of driest month	
Bio15	Precipitation seasonality (coefficient of variation)	
Bio16	Precipitation of the wettest quarter	
Bio17	Precipitation of driest quarter	
Bio18	Precipitation of warmest quarter	
Bio19	Precipitation of coldest quarter	

We used the same bioclimatic variables projected into the future. Potential values for bioclimatic variables for future climate conditions in the 2020s, 2050s and 2080s, respectively, were derived from three general circulation models (GCMs): BCCR-BCM2.0 (BC), CCCMA_CGCM3 (CC), and MIROC32_medres (MI), under for the SRES-A1B, SRES-A2 and SRES-B1 emission scenarios (IPCC, 2001).The SRES was a report by the IPCC which was published in 2000. The SRES was a set of emissions scenarios. It covers a wide range of the main driving forces of future emissions, from demographic to technological and economic developments (IPCC, 2001). SRES-A1B means a balanced across energy sources in future world. The SRES-A2 driven by economic growth although at regional scale, create a diversified political and social world and regionally oriented economic growth that is more fragmented and slower than that in other storylines. The SRES-B1 is undoubtedly the “greenest” of all projections, with high levels of environmental and social consciousness and a global sustained development (IPCC, 2001).

The bioclimatic variables for the future scenarios were from the IPCC Fourth Assessment Report (available from the International Centre for Tropical Agriculture: http://ccafs-climate.org). This method assumes changes in climates only over large distances and the relationships between variables are maintained from the present towards the future (see http://ccafs-climate.org/ for more details). Future climate variables also had a resolution of 30” (about 1 km2).

Modeling procedures and testing

Based on the selection criteria of the standardization method, perfect results could be obtained by using the membership function in the fuzzy mathematics (Lu et al., 2012; Guo et al., 2013). For defining the degree of suitability of all factors in a finite set (scope) for S. sphenanthera, the values of the fuzzy membership using the statistical association between the content of schisantherin A and each factor was used to quantify the relation between habitat variables and habitat suitability. For each environmental variable, a suitability curve is defined that assigns every possible value of the variable a degree of suitability. The degree value ranges from 0 to 1. A value of 0 indicates that an attribute has complete non-membership (not suitable at all) in a fuzzy set, which means that under these environmental conditions, S. sphenanthera cannot grow or cannot completely synthesize and accumulate schisantherin A. Meanwhile, a value of 1 indicates that an attribute has a complete membership (optimal suitable) in a fuzzy set, which means that under these environmental conditions, the synthesis and accumulation of schisantherin A in S. sphenanthera is optimal, and the environment in this scenario is suitable for the growth of S. sphenanthera. Grades between 0 and 1 imply that an attribute has partial of membership in a fuzzy set. (Rüger, Schlüter & Matthies, 2005; Mouton et al., 2011).

In this study, 75 % of the distribution points were selected at random as the training data set, and the remaining points were used as the test data set. Among various existing membership functions, the Gaussian membership function (Eq. (1)) and the Trapezoidal membership function (Eqs. (2) and (3)) were used to standardize each factor (Table 2) (Lahdelma, Salminen & Kuula, 2003). The error inspection of each factor to the membership function was computed by Matlab 2011 (MathWorks, 2011). (1) Ax=e−x−aσ2,−∞<x<∞

where x is the independent variable, and a, σ are parameters, in this definition when x = a, the membership value is 1. (2) Ax=1,x<ab−xx−a,a≤x≤b0,b<x

(3) Ax=0,x<ax−ab−a,a≤x≤b1,b<x

In Eqs. (2) and (3), where x is the independent variable, and a, b are parameters.

Table 2 Membership functions of factors and its fitting degree.

Code (Unit)	Membership function	Optimal value	Threshold value	Fitting degree	Weight (%)	
Bio1 ( °C)	Gaussian-type	10.3	7.3, 13.2	85.44	6.37	
Bio2 ( °C)	Subsection Gaussian-type	88.4	70, 135	85.70	5.48	
bio3	Gaussian-type	0.26	0.22, 0.31	80.52	3.74	
bio4	Gaussian-type	83.10	72.16, 4.03	80.36	4.13	
Bio5 ( °C)	L trapezoidal	>26.70	25.20	89.02	3.33	
Bio6 ( °C)	L trapezoidal	<−2.70	−8.60	83.50	3.74	
Bio7 ( °C)	Gaussian-type	33.00	30.10, 35.80	80.81	7.10	
Bio8 ( °C)	R trapezoid	23.00	18.10	93.40	5.21	
Bio9 ( °C)	Gaussian-type	5.60	−20.00, 32.00	79.54	1.34	
Bio10 ( °C)	R trapezoid	23.80	19.20	96.93	5.36	
Bio11 ( °C)	Gaussian-type	4.04	−21, 29	76.63	5.31	
Bio12 (mm)	Gaussian-type	853.00	525.00, 1181.00	77.34	6.51	
Bio13 (mm)	Gaussian-type	144.00	96.00, 192.00	86.52	5.92	
Bio14 (mm)	Gaussian-type	10.14	0.94, 19.00	81.27	5.49	
Bio15	Gaussian-type	0.70	0.56, 0.82	84.27	5.82	
Bio16 (mm)	Gaussian-type	415.20	294.00, 536.00	84.97	6.79	
Bio17 (mm)	Gaussian-type	36.11	7.17, 65.04	78.75	5.76	
Bio18 (mm)	Gaussian-type	367.20	251.00, 483.00	76.20	6.66	
Bio19 (mm)	Subsection Gaussian-type	27.77	5.36	96.60	5.94	

After establishing the factor’s fuzzy membership for S. sphenanthera, the suitability ratings were calculated. To do so, assigning the weight of different factors is very important. This is the key to the reliability of the evaluation results. The factor weight is the contribution degree of the effects of factors to the growth and quality of S. sphenanthera, which varies among different factors. To eliminate subjective influence, the current study takes advantage of the maximum entropy method (MEM) to determine the weights of factors. MEM is a parameter that describes how many different alternative approaches exist with respect to a certain attribute. Maximum information entropy weighting models embody the observed characteristics of the data, and the greatest degree of avoiding the influence of artificial factors is achieved. At the same time, according to the degree of change in the data set, the multidimensional ecological dominant factors that affect the species distribution in space are identified. Additionally, it does not require much data, which are rarely available (Phillips, Anderson & Schapire, 2006) can produce solutions with a small set of observations (Weber, 2011).

Based on the MEM and the fuzzy mathematics, membership functions were calculated to obtain the values of all factors and units, while the weights were obtained from the MEM. Then, the weighted average method [Eq. (4)] was used to comprehensively assess the values of S. sphenanthera ecological suitability in the study area (Lu et al., 2012; Guo et al., 2013). (4) yi= ∑j=1nwj×xij

where yi (integrated ecological suitability index) refers to the comprehensive value of assessment of S. sphenanthera ecological suitability in each evaluation unit, n is the number of factors, wj is the weight of a factor, and xij is the value of the ith grid cell for the jth variable. The scope of yi was defined as 0–1. The higher the yi value means the greater the contribution of ecological factors to the quality level of S. sphenanthera. We applied spatial analysis using ArcGIS 9.3 (Redlands, CA, USA) to calculate the raster and to set the cell size at the maximum of the inputs. Therefore, the maps of the habitat have a suitability assessment resolution of 30” (about 1 km2).

The root mean square error (RMSE) and determination coefficient (R-squared) were used to validate the results of the habitat suitability assessment for this plant. RMSE and R-squared were calculated as (Eq. (5)) (Willmott, 1984) and (Eq. (6)) (Magee, 1990): (5) RMSE=1N∑i=1NPi−Qi2

(6) R2=1−∑i=1NQi−Pi2 ∑i=1NQi−Q¯2

where N is the number of samples, Pi is the ith predicted value and Qi represents the ith observed value and Q¯ is the mean of the observed value. The smaller the RMSE is, the better the interpolation method (Li & Heap, 2011). R-squared summarize the discrepancy between observed values and their predicted values, the values of R-squared vary from 0 to 1. Meanwhile, to test the applicability of our model in other habitats besides the Qinling Mountains, 60 samples from four locations (Fig. 1, 1–4) were collected, and the data were assessed using the above mentioned habitat suitability assessment model.

Potential change in the distributions of species

We used the above mentioned model to study the potential changes in the distributions of S. sphenanthera. The future scenarios were regionalized for the study area from three general circulation models (GCM) for the SRES-A1B, SRES-A2 and SRES-B1 (IPCC, 2001), and for the three time (2020s, 2050s and 2080s) periods. All operations were automated with a script made in python programming language and incorporated as a toolbox in ArcGIS 9.3 (Redlands, CA, USA).

Results

The determination of the fuzzy membership functions

The membership function was established to explore the relationships between factors and membership degrees by cure-fitting and drawing scatter diagrams. Because the interaction between the growth and quality of S. sphenanthera and ecological factors was quite different, different types of fuzzy membership functions were chosen to express their relationships. Therefore, in this research, gaussian-type functions were used for quantitative factors because most factors have upper and lower boundaries in which the membership degree is zero. The L trapezoidal function was used for quantitative factors that only had lower boundaries, and the R trapezoid type function was used for quantitative factors that only had upper boundaries. The membership degrees, extending from 0 to 1, can be calculated according to these membership functions. When the membership degree is 1, it is considered to be the best scenario of ecological factors suitable for the growth of S. sphenanthera. However, when the membership degree is 0, the ecological factor has a serious deficiency. The fitting degree of all membership functions varies from 76.20 % to 96.93 % (Table 2), with an average of 84.09 %. This verifies our model’s accuracy.

Figure 2 Habitat suitability comprehensive distribution of S. sphenanthera in Qinling area.

Distribution of suitable habitats in the current climate environment

The Chinese Pharmacopoeia clearly states that the minimum content of schisantherin A in S. sphenanthera fruit is 0.2 % (Committee of National Pharmacopoeia, 2010). In this study, the content of schisantherin A in the 307 samples was 0.015–1.948 %, we designated the content proportions of schisantherin A less than 0.2 % as unsuitable. According to the content of schisantherin A and the ecological suitability index calculated by our model, the habitat suitability of S. sphenanthera in the Qinling Mountains can be classified into four grades: unsuitable, marginally suitable, moderately suitable, and highly suitable (yi < 0.3, unsuitable habitat; 0.3 ≤ yi < 0.5, marginally suitable habitat; 0.5 ≤yi < 0.7, moderately suitable habitat; and yi ≥ 0.7, highly suitable habitat). We have applied spatial analysis using ArcGIS 9.3 (Redlands, CA, USA) to draw the comprehensive distribution map of habitat suitability of S. sphenanthera in the Qinling area (Fig. 2), and we calculated the area of all of the habitat suitability grades. The results show that the proportions of unsuitable, marginally suitable, moderately suitable, and highly suitable habitats for S. sphenanthera in the Qinling Mountains are 1.12 %, 11.76 %, 43.69 % and 43.43 %, respectively. The highly suitable habitats are mainly located in southern Shaanxi, eastern Gansu, western Henan, and northern Hubei province, and their surrounding areas, with an elevation of 800 to 2000 m.

The RMSE of the habitat suitability assessment for S. sphenanthera is 0.056 and the R-squared is 0.76, indicating that the simulation values are highly consistent with the observed values. Therefore, the model was built in the study is accurate and available. Moreover, the RMSE is 0.075 and the R-squared is 0.68 for the data from the 60 sample sites outside of the Qinling Mountains district, which demonstrates that the model is applicable to other areas besides the Qinling Mountains.

Projecting the effects of climate change on the distribution and quality of S. sphenanthera in the Qinling Mountains

The habitat suitability assessment model based on GIS and the fuzzy mathematics could be used to accurately determine the habitat suitability of S. sphenanthera and to quantify the area of suitable habitat. Using it as a base, the future distributions of S. sphenanthera in the 2020s, 2050s and 2080s under the climate change scenarios of SRES-A1B (Fig. 3), SRES-A2 (Fig. 4) and SRES-B1 (Fig. 5) from IPCC were projected. The SRES scenarios were constructed to explore future developments in the global environment with special reference to the production of greenhouse gases and aerosol precursor emissions, which is an important means of evaluating the future climate change.

Figure 3 Distribution map of habitat acclimatization for S. sphenanthera based on three GCM in SRES-A1B.

(A) 2020s in GCM of BC; (B) 2050s in GCM of BC; (C) 2080s in GCM of BC; (D) 2020s in GCM of CC; (E) 2050s in GCM of CC; (F) 2080s in GCM of CC; (G) 2020s in GCM of MI; (H) 2050s in GCM of MI; (I) 2080s in GCM of MI. Future scenarios were regionalized within the study area from the following general circulation models (GCM): BCCR-BCM2.0 (BC), CCCMA_CGCM3 (CC), and MIROC32_medres (MI).

Figure 4 Distribution map of habitat acclimatization for S. sphenanthera based on three GCM in SRES-A2.

(A) 2020s in GCM of BC; (B) 2050s in GCM of BC; (C) 2080s in GCM of BC; (D) 2020s in GCM of CC; (E) 2050s in GCM of CC; (F) 2080s in GCM of CC; (G) 2020s in GCM of MI; (H) 2050s in GCM of MI; (I) 2080s in GCM of MI. Future scenarios were regionalized within the study area from the following general circulation models (GCM): BCCR-BCM2.0 (BC), CCCMA_CGCM3 (CC), and MIROC32_medres (MI).

Figure 5 Distribution map of habitat acclimatization for S. sphenanthera based on three GCM in SRES-B1.

(A) 2020s in GCM of BC; (B) 2050s in GCM of BC; (C) 2080s in GCM of BC; (D) 2020s in GCM of CC; (E) 2050s in GCM of CC; (F) 2080s in GCM of CC; (G) 2020s in GCM of MI; (H) 2050s in GCM of MI; (I) 2080s in GCM of MI. Future scenarios were regionalized within the study area from the following general circulation models (GCM): BCCR-BCM2.0 (BC), CCCMA_CGCM3 (CC), and MIROC32_medres (MI).

Using the projected climate data for the studied area, we reassessed the current S. sphenanthera distribution to reflect a changing climate. These predictions provide a comprehensive assessment of S. sphenanthera ecological suitability. Using the same standard of classification, the future habitat suitability of S. sphenanthera in the Qinling Mountains is classified into four grades. The data were derived from the average of the three types of GCMs (Fig. 6). The model forecast indicates that in the study area, the unsuitable habitats of S. sphenanthera under climate change scenarios SRES-A1B and SRES-A2 were gradually increased. As time goes on, unsuitable habitat areas will expand in the northeast of Chongqing and the northwest of Hubei (Figs. 3 and 4). The unsuitable habitat of SRES-B1 does not represent a large change, even a slight reduction with time. The marginally suitable habitat of S. sphenanthera under the three climate change scenarios would rapidly increase. The difference between the climate changes scenarios is a higher increasing rate in the SRES-A1B and SRES-A2 scenarios. The moderately suitable habitat of S. sphenanthera under all climate change scenarios remained relatively stable. The highly suitable habitat of S. sphenanthera under the climate change scenarios of SRES-A1B, SRES-A2 and SRES-B1 gradually decreased. Some of the differences between the climate change scenarios are worth mentioning. Generally, in the study area, a larger decline rate of highly suitable habitat area in both GCMs refers to the SRES-A1B and SRES-A2.

Figure 6 The variation of the percentage of areas of habitat suitability comprehensive distribution of S. sphenanthera under different climate change scenarios.

(A) SRES-A1B; (B) SRES-A2; (C) SRES-B1.

Climate change will significantly affect plant distribution. With the support of the ArcGIS 9.3 (Redlands, CA, USA) platform, we carried out statistics that can represent the future climate data in the model operations. To explore the relationships between climatic change and the distribution of S. sphenanthera, SPSS 19.0 software (IBM, Armonk, North Castle, NY, USA) was used to determine the correlations between Bio1 (annual mean temperature), Bio12 (annual precipitation) and the area proportion of the four habitat types of S. sphenanthera in the studied area. Table 3 shows that Bio1, Bio12 and the area of highly suitable habitat have a significant negative correlation. As Bio1 and Bio12 increased, the highly suitable habitat area gradually decreased. To further explore this quantitative relation and future change trend, we performed linear regression analysis (Fig. 7). The results indicate that when Bio1 of the studied area exceeds 20 °C and Bio12 exceeds 1200 mm, the study area will no longer have highly suitable habitat areas for S. sphenanthera. The result showed that climate change will affect the distribution and quality of S. sphenanthera, which will further increase the rate of this species’ extinction.

Table 3 The correlation between Bio1, Bio 12 and S. sphenanthera’s different grades of habitat suitability in the studied area.

Code	Unsuitable habitat	Marginally suitable habitat	Moderately suitable habitat	Highly suitable habitat	
Bio1	.151	.380*	.292	−.514**	
Bio12	.326	.558**	.051	−.598**	
Notes.

* Correlation is significant at the 0.05 level (2-tailed),

** Correlation is significant at the 0.01 level (2-tailed).

Figure 7 Linear regression analysis of Bio1, Bio12 and its corresponding percentage of highly suitable habitats of S. sphenanthera.

(A) Bio1 (Annual mean temperature); (B) Bio12 (Annual precipitation).

Discussion

Advantages of the methods and implications for ecological modeling

The reliability of the evaluation results depends on the integrity of the database and the rationality of the evaluation methods (Lu et al., 2012; Guo et al., 2013). GIS is an important tool for the analysis of medicinal plants’ ecological suitability using species presence records associated with environmental variables to determine the essential environmental requirements of a particular species. The suitability evaluation model was integrated with the GIS software ArcGIS, which could quickly provide evaluation results after inputting the parameters of the model. Fuzzy theory was widely used in ecological environment modeling (Rocchini, 2014; Mouton et al., 2011), and how to determine membership function is a crux of this theory. Usually, the parameters of fuzzy membership functions were calculated based on the expert knowledge (Zhu et al., 2010; Wang, Hong & Tseng, 2000). But in this study, to make sure the consistency between the sampling data and the bioclimatic data, we extract bioclimatic variables values by sampling sites, and then get the bioclimatic data sequence for model training. Finally, using this data, the fuzzy membership function was defined, which include the information on relationship between the value of 19 climatic variables and schisantherin A content of S. sphenanthera samples. This method can standardize variables with no prior knowledge as well as retain information about the original data.

There are many methods for determining the weight of factors, such as the Delphi (Lehtonen & Tykkyläinen, 2014), AHP (Houshyara et al., 2014) and principal component analysis (Alfaro et al., 2014). The methods such as Delphi and AHP need priori knowledge or additional expertise, but S. sphenanthera is a wildlife species, and is lacking sufficient expert knowledge and conventional established definitions. For principal component analysis, to get reasonable results, the model requires a lot of data, so it is not suitable for this research. MEM is an objective approach using information provided by the observation data to determine the weight of climatic variables. It can objectively evaluate the habitat suitability and eliminate subjective influence. MEM emphasizes the influence of strongly restrictive factors on targets. Moreover it determines the dominant influencing factor of schisantherin A.

Habitat requirements and spatial distribution of species are essential to biological conservation and management. In this study, our aim is to establish a universal model for simulation quality of wild herbs in different environmental conditions, which means For wild medical plant without explicit habitat requirements, this study can only be use the information provided by the statistical data to build up a model, and according to the research target, this model should not only forecast spatial variation of medicinal plants quality, but also can determine the mathematical relationship between value of each impact factor and the content of effective composition in wild herbs. Therefore, the general statistical model cannot meet the requirements of this study. Moreover, the samples data in this study were collected by field work in Qinling Mountains, every data consumed extremely labor, material, and financial resources, especially in mountains or regions of complex terrain. Thus, only limited data was provided for model building, it can’t meet the requirements of other machine learning model such as artificial neural network model, support vector machine, and random forest. At the same time, most of the machine learning models are a black-box model, which cannot show relationship between the value of each impact factor and the content of effective composition. This means machine learning models are also not suitable for this study. However, Fuzzy mathematics provides a way to solve this problem. Firstly, fuzzy membership function can standardize the factors with no prior knowledge and retain the information of original data. Secondly, through this method we can obtain suitable range of each factor for medical plant, and this information is significant for wildlife protection. Finally, this method does not require large amounts of data, and gives a satisfactory result.

The reasonability of the bioclimatic variables

Bioclimatic variables are derived from the monthly temperature and rainfall values in order to generate more biologically meaningful variables. These variables were created by the thin-plate smoothing spline algorithm implemented for interpolation, using latitude, longitude, and elevation as independent variables, and the uncertainty arising from the input data and the interpolation was quantified by mapping weather station density, elevation bias in the weather stations, and elevation variation within grid cells and through data partitioning and cross validation. All of this guarantees the accuracy of the bioclimatic data (Hijmans et al., 2005). Furthermore, those data had been proven to be useful for SDMs; at the same time, it’s also one of the most widely used data (Rödder & Lötters, 2009).

Previous researches indicated that temperature and the moisture factor (especially in spring and autumn) are the main climate factors which control the growth of plants (Trisurat, Shrestha & Kjelgren, 2011). Only under the circumstances of appropriate temperatures and sufficient water can plants have normal growth and accumulate particular organics. S. sphenanthera is a woody vine, supporting itself on rocks, shrubs, and trees in broad-leaved evergreen forests or coniferous mixed forests (Committee of Flora of China, 1996). On the base of phenological observation, we have a preliminary understanding of the mechanism of the S. spenanthera. In S. sphenanthera growth process, the period of sap flow begin in early March, and the sprout leaves period is from late March to mid-April, in this period, low temperature or low rainfall will kill the buds or delay sprout leaves period. In late April S. spenanthera begin to blossom, and the blossom and young fruit period continue to end of June, at the same time it’s also a peak-growth period. The developing fruit period is begin in early July, and the fruit ripe period begin in early August, then its followed by abscission period which last from October to November, and from July to September is the warmest quarter in study area and usually, the wettest month also happen in this three months, it is the key period for generation and accumulation of schisantherin A. Therefore, in the whole growth period, suitable temperature and precipitation is significant for S. sphenanthera. The bioclimatic variables represent annual trends, seasonality, and extreme or limiting temperature and precipitation factors in plant life history, so they are sufficient for this research.

Lu et al. (2012) found that temperature and precipitation in the growing period have important effects on the distribution of S. sphenanthera. Hu et al. (2012) found that, according to MaxEnt, Bio13 (precipitation of the wettest month), Bio18 (precipitation of the warmest quarter), and Bio1 (annual mean temperature) have important effects on the distribution of Schisandra chinensis. S. chinensis, another species in the genus. In this study, according to the contributions of individual factors to the content of schisantherin A (Table 2), five dominant climatic factors controlled S. sphenanthera distribution. These five dominant features are Bio7 (temperature annual range), Bio16 (precipitation of the wettest quarter), Bio18 (precipitation of the warmest quarter), Bio12 (annual precipitation) and Bio1 (annual mean temperature). The weight of each of these features is higher than 6 %. Our results are therefore somewhat similar with those of Lu et al. (2012) and Hu et al. (2012).

The effects of climate change on the highly suitable habitat of S. sphenanthera

Over time, a new suitable habitat for S. sphenanthera would be generated while its normally suitable habitat would be lost. To show this change, we expanded the areas around the studied area to generate the future suitable habitat of S. sphenanthera. The highly suitable habitat of S. sphenanthera under the climate change scenarios of SRES-A1B and SRES-A2 was shifted to the north and gradually reduced (Figs. 3 and 4). By the 2020s, the highly suitable habitat area in eastern Gansu would be significantly reduced. The same situation would occur in Shaanxi, Hubei and Henan provinces with habitat fragmentation in the study area (Figs. 3A, 3D, 3G, 4A, 4D and 4G). By the 2050s, there would be a highly suitable habitat in the east of Gansu, which is located in the north of the studied area. There was little or almost no highly suitable habitat in Hubei. The highly suitable habitats of S. sphenanthera would continue to reduce in Shaanxi and Henan, with further habitat fragmentation occurring in the study area. In the meantime, outside of the studied area, new highly suitable habitats are likely to appear in northern Gansu, central Shaanxi and central Shanxi (Figs. 3B, 3E, 3H, 4B, 4E and 4H). By the 2080s, in the study area, the highly suitable habitat of S. sphenanthera would fall sharply and only a little area in parts of southern Shaanxi and west Henan would remain. However, outside of the study area in northern Gansu, central Shaanxi and central Shanxi, new highly suitable habitat areas would be further expanded, leading to a new distribution area of S. sphenanthera (Figs. 3C, 3F, 3I, 4C, 4F and 4I).

The other factors which affect the distribution of S. sphenanthera

In addition to climate change, human interference is also an important factor in influencing the distribution of S. sphenanthera. Theoretically, S. sphenanthera can grow in all of the highly suitable habitats within the scope of its distribution area. Due to the effects of human activities such as over-grazing and wanton picking, wild S. sphenanthera resources are facing depletion. Traditional habitats no longer host this species. The phenomenon of habitat fragmentation is irreversible. The proliferation ability of S. sphenanthera is also a key factor affecting its distribution. Because S. sphenanthera is a woody vine, its growth has more stringent environmental requirements. Research has shown that land cover types in the highly suitable habitats of S. sphenanthera include broad-leaved evergreen forests that provide support to climbing plants (Lu et al., 2012). Future climate warming will impact the highly suitable habitats of S. sphenanthera. Additionally, the distribution of the highly suitable habitats of S. sphenanthera would shift to higher latitudes in response to climate change. However, in the studied area, like northern Shaanxi and some parts of the eastern Gansu area, the present vegetation type is not suitable for S. sphenanthera growth. At present, because the understanding level and the technical conditions are limiting, this study cannot include all factors influencing the distribution and quality of S. sphenanthera. If all factors were considered, we can deduce that in the study area, the suitable habitat of S. sphenanthera under the climate change scenarios would reduce by a larger range, and new suitable habitat range will be smaller.

Conclusions

Medicinal plants represent a significant contribution to human health, and with the continuous development of global warming, some of wild medicinal plants would change their distribution patterns as well as quality. This study provided a new case for that theory. Firstly, samples of wild herb plant S. sphenanthera were collected form field work in Qinling Mountains, and the content of schisantherin A in the fruit were examined by HPLC, then with this data combining fuzzy theory and a maximum entropy model, we developed a method to project the spatial assessment of medicinal plants quality, and used this model we predict future quality and distribution of S. spenanthera in Qinling Mountains. The result showed that under future climate scenarios, the habitats of high quality S. sphenanthera will continue to decrease and draw near to extinction. As the prediction of species distribution is influenced by multiple factors, this study predicted the effects of climate change on the distribution and quality of medicinal plants. Further research is required, and the present study only provides a reference for related research and native habitat protection.

Supplemental Information

Supplemental Information 1 Fruits contents of Schisantherin A in sample of Schisandra sphenanthera (%)

Click here for additional data file.

Additional Information and Declarations

Competing Interests

Author Contributions

Data Availability

The authors declare there are no competing interests

Yanlong Guo conceived and designed the experiments, performed the experiments, analyzed the data, contributed reagents/materials/analysis tools, wrote the paper, prepared figures and/or tables, reviewed drafts of the paper.

Haiyan Wei conceived and designed the experiments, performed the experiments, analyzed the data, reviewed drafts of the paper.

Chunyan Lu reviewed drafts of the paper, have made great contributions to revision.

Bei Gao analyzed the data, contributed reagents/materials/analysis tools, prepared figures and/or tables, reviewed drafts of the paper.

Wei Gu conceived and designed the experiments, performed the experiments, contributed reagents/materials/analysis tools, reviewed drafts of the paper.

The following information was supplied regarding data availability:

The raw data has been supplied as a Supplemental File.

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
