# Peer review of "Predictions of potential geographical distribution and quality of Schisandra sphenanthera under climate change"

_PeerJ, doi:10.7717/peerj.2554_

## Round 0.1 · original submission · Major Revisions

Based on the reviews, and my own evaluation of the manuscript, I have determined that this article requires substantial revision; therefore I cannot accept this version of the manuscript for publication. However, I hope that you will be willing to submit a suitably revised version.

·

Basic reporting

The language is OK, but still Some clumsy sentences or many mistakes exsit in the expressions or grammer. E.g. line19-21, line25-26, line105,119-124 and so on. Pay attention to the using of punctuation.
The structure of the submitted article should conform to one of the templates. At least ,a section of CONCLUSION should be added to draw a conclusion for your research.

Experimental design

The advancement of the research method need to be further discussed. It is feasible that the “fuzzy theory” and maximum entropy model are utilized to forecast the distribution of S. spenanthera, but there are more advanced and suitable methods or models to settle the problem, for example, artificial neural network model, Support Vector Machine and so on. The author should explain that the methods utilized are better than other models.
This paper did not analyze the mechanism of the S. spenanthera. As we all know, in the study of the predictions to geographical distribution, the mechanism is very important because it can affect the choice of data and methods. From the statement of this paper, I am puzzled that how the S. spenanthera grows. I wonder if bioclimatic data were sufficient for this research because that the topography and landform factor, geology factor, etc. also influence the growth of S. spenanthera.
The spatial distribution of the sampling points is vague. For the geographical distribution prediction of S. spenanthera, the rationality of sampling point distribution is vital because that it affects the validity of the prediction results based on the interpolation of those sample point.

Validity of the findings

The accuracy of the bioclimatic data is not explained. The consistency of the sampling data and the bioclimatic data is important because it affected the accuracy and precision the modeling.

Additional comments

This paper use “fuzzy theory” and maximum entropy model to estimate the potential changes in quality and distribution of S. spenanthera, and its result will be very instructive in the S. spenanthera conservation and management. However, I think that this paper cannot be accepted at the present state. Some questions should be answered before. The language should be further polished. Some sentence should be more cautiously expressed, like "..., none heve incorporated the ..." in the abstract. Survival mechanism of S. spenanthera should be introduced to show why data and methods were selected in this paper. And what’s the merits of your method? For the structure, there is no CONCLUSION part!

·

Basic reporting

No Comments

Experimental design

No Comments

Validity of the findings

No Comments

Additional comments

In the manuscript of “Predictions of potential geographical distribution and quality of Schisandra sphenanthera under climate change”, authors used a fuzzy theory model combined with General Circulation Model outputs to analyze the quality and distribution of Schisandra sphenanthera (S. sphenanthera) in both historical and future scenarios. The contributions of this manuscript are (1) the investigation of the impacts of climate changes on an endangered trandional Chinese medical plant and (2) the provision of vegetation management for S. sphenanthera. The manuscript is well written and should be interesting to policy makers and researchers in fields of ecosystem and traditional Chinese medicine. With regards to the methodology and experiments, there are nothing major in reviewer’s opinion. And figures and equations are appropriate. Here, I listed some minor issues and suggestions for further improvement.
1. Authors use the maximum entropy method in obtaining the optimal weight for each decision variable. Reviewer is wondering the consistency of the optimal weights. A comparison of the weights derived from MEM is suggested to compare with other methods, maybe Gini diversity index? Which is another diversity index used in ecological studies another other many fields.
2. In the evaluation of results of the habitat suitability assessment, only RMSE is used. Other statistical measures are also suggested to be included.
3. Are the approaches used in this study have the suitability and applicability for other plants?
4. Any limitation of using the fuzzy theory model and the MEM?
5. There are a couple long sentences that hard to understand, such as line 121-124. Authors should check all content and make sure that any long sentence is readable. If not, the use of a couple short sentences are suggested.
6. There are some English grammar errors identified by reviewer, such as the missing “the” before “Qinling Mountain”, the missing word “a” before fuzzy theory and MEM and others, the improper use of “comma” instead of a full stop (“period”) in line 218 after the percentage.

---

## Round 0.2 · Minor Revisions

I read carefully this updated version of the manuscript and the related rebuttal. The authors had addressed most of the comments pointed out by the reviewers. Hence, the conditional acceptance (minor revision) is the determination. Please address the remaining comments.

·

Basic reporting

No Comments

Experimental design

No Comments

Validity of the findings

No Comments

Additional comments

1. The language of this paper is more fluent, and the use of punctuation is more reasonable.
2. The author makes a detailed description of the reason for using fuzzy theory, but I think that the author does not fully explain that the fuzzy theory is the most suitable method for the research object of this paper. My suggestion is that the author should elaborate the relation between the research object and its influencing factor. On the basis of this, the fuzzy theory is the most suitable research methods, or by comparison with other methods (such as artificial neural network model, Support Vector Machine and so on), the paper illustrate the fuzzy theory is better comparing to the other methods.
3. I don’t think the author has completely answered my question. It is correct that the paper studies the predictions of potential geographical distribution and quality of S. sphenanthera under climate change, but if the potential geographical distribution of S. sphenanthera is also influenced by the topography and landform factor, geology factor, etc, I think that result of predictions of potential geographical distribution and quality is not persuasive only considering the factors of climate change.
4. The author fully explains how to choose the sampling points for the study.
5. The author fully explains the method and accuracy for obtaining the bioclimatic data.
6. The sentences of the paper are more cautiously expressed through the author’s careful modification.
7. The conclusion of the modified paper is sufficient and reasonable.

·

Basic reporting

In this round of revision, the authors are able to address most of the issues raised by reviewer. With regard to this version, reviewer has no objection of publication after journal editorial editing.

Experimental design

Experiments are well done and design is proper to support authors' conclusion.

Validity of the findings

The findings are significant with respect to Schisandra sphenanthera growth in China under varying climate conditions.

---

## Round 0.3 · accepted · Accept

I am pleased to say that the revised version of the manuscript is suitable for publication